# A Straightforward Bifurcation Pattern-Based Fundus Image Registration Method

**DOI:** 10.3390/s23187809

**Published:** 2023-09-11

**Authors:** Jesús Eduardo Ochoa-Astorga, Linni Wang, Weiwei Du, Yahui Peng

**Affiliations:** 1Information and Human Science, Kyoto Institute of Technology University, Kyoto 6068585, Japan; d1821501@edu.kit.ac.jp; 2Retina & Neuron-Ophthalmology, Tianjin Medical University Eye Hospital, Tianjin 300084, China; 3School of Electronic and Information Engineering, Beijing Jiaotong University, Beijing 100044, China; yhpeng@bjtu.edu.cn

**Keywords:** image registration, fundus image, retinal analysis, blood vessel bifurcation, registration error

## Abstract

Fundus image registration is crucial in eye disease examination, as it enables the alignment of overlapping fundus images, facilitating a comprehensive assessment of conditions like diabetic retinopathy, where a single image’s limited field of view might be insufficient. By combining multiple images, the field of view for retinal analysis is extended, and resolution is enhanced through super-resolution imaging. Moreover, this method facilitates patient follow-up through longitudinal studies. This paper proposes a straightforward method for fundus image registration based on bifurcations, which serve as prominent landmarks. The approach aims to establish a baseline for fundus image registration using these landmarks as feature points, addressing the current challenge of validation in this field. The proposed approach involves the use of a robust vascular tree segmentation method to detect feature points within a specified range. The method involves coarse vessel segmentation to analyze patterns in the skeleton of the segmentation foreground, followed by feature description based on the generation of a histogram of oriented gradients and determination of image relation through a transformation matrix. Image blending produces a seamless registered image. Evaluation on the FIRE dataset using registration error as the key parameter for accuracy demonstrates the method’s effectiveness. The results show the superior performance of the proposed method compared to other techniques using vessel-based feature extraction or partially based on SURF, achieving an area under the curve of 0.526 for the entire FIRE dataset.

## 1. Introduction

The human body comprises numerous systems for its functioning; however, the visual system stands out as one of the most important. Malfunctions within this system can be caused by various factors, and in severe cases, they can lead to vision loss. Given that most diseases causing complications manifest in the retina, the analysis of this structure is relevant. Consequently, retinal analysis techniques have become a fully consolidated method for analysis and classification of several retinal diseases, including diabetic retinopathy [1,2,3,4], macular degeneration [5,6,7,8], and glaucoma [9,10,11,12], among others. Among the various diagnostic tools used for this purpose, fundus photography stands out as the most common method for screening of certain retinal diseases.

However, retinal analysis in fundus photography typically remains confined to a narrow scope, encompassing less than 50 degrees [13,14] and including the optic disc and fovea. Consequently, this limited field of view poses challenges in achieving a comprehensive analysis of the entire retina. To address this issue, physicians often resort to analyzing various images from different perspectives separately, which can be confusing and time-consuming [15,16]. While wide-angle fundus photography offers a promising alternative to obtain a broader view of the retina, practical limitations hinder its widespread adoption, especially for patients in rural areas [17]. Figure 1 shows a comparison between a conventional fundus image and a wide-angle fundus image.

Image registration plays a crucial role in longitudinal studies by aligning fundus photographs of the same region at different times. This process enables the detection of morphological changes in the retina, facilitating follow-up of the patient’s condition during treatment or evaluation of the progression of a disease. Similarly, registration is also valuable in remote screening and is utilized as part of super-resolution imaging [19]. By employing multiple images captured from slightly different angles, this technique enhances the clarity of clinical features, providing valuable insights for accurate diagnoses.

The image registration process involves aligning a fixed or target image to a moving or source image. Two main approaches are commonly employed: intensity-based and feature-based techniques. The intensity-based approach determines the warping parameters by minimizing a function that measures the disparity in intensities between images. While this method can yield successful results in certain cases, it is not recommended for scenarios with significant illumination changes, variations in texture, or content differences, as often encountered in longitudinal studies.

On the other hand, feature-based approaches rely on detecting distinctive landmarks in the images, which serve as reference points for relating and aligning the images, thus estimating the warping parameters. This approach is particularly beneficial when dealing with images that require substantial displacements to be accurately registered. Presently, feature-based techniques dominate the field of fundus image registration due to their robustness and flexibility [20].

The challenges in fundus image registration encompass image variability caused by exposure or field-of-view disparities, pathological abnormalities altering retinal morphology, and obstructions in structures. Moreover, the scarce ground truth hinders the quantitative evaluation of registration methods. Considering this, a dataset with ground truth is utilized to assess a feature-based fundus registration method. This approach incorporates a robust vascular tree segmentation method to identify feature points within a defined scope. The method is also tested on registrable image pairs with pathological changes, and results are presented based on registration error, a crucial factor in determining accuracy. Main results indicate that, despite its relative simplicity, the method demonstrates performance similar to that of other techniques. One strength of the approach lies in the distribution of feature points across the entire fundus image. Unlike more general-purpose feature detectors like SIFT, SURF, or ORB, the method guarantees the dispersion of feature points throughout the image, potentially leading to enhanced accuracy in registration outcomes.

While image registration for medical purposes has been a research topic for several decades, some works related to fundus image registration still rely on qualitative assessment due to the lack of ground truth. As validation is a key factor for its substantial inclusion in medical practice, it is essential to perform a quantitative evaluation of the performance of fundus registration methods.

Proposing this feature-based fundus image registration method, challenges in retinal analysis are addressed, including limited field of view and accurate longitudinal studies. This approach utilizes vascular tree segmentation to pinpoint feature points, ensuring image alignment. Notably, the method’s favorability lies in the uniform distribution of points across the image, enhancing registration accuracy. Quantitatively evaluated by registration error, this technique matches the performance of complex alternatives while offering simplicity and clinical potential.

This paper is organized as follows, Section 2 presents an overview of related work in the field. Subsequently, Section 3 outlines the proposed method for fundus image registration. In Section 4, the experimental results obtained by applying the proposed technique are reviewed. In Section 5, a discussion of the results is presented. Finally, Section 6 concludes the paper.

## 2. Related Work

Several researchers have explored different approaches for fundus image registration, aiming to align retinal images accurately. In the work by Catin et al. [21], a retina mosaicking method was introduced using a feature point detector based on a Hessian matrix approach and a 128-dimensional distribution to describe the surrounding region of the feature point. Their approach sought independence from the presence of vascular structures in the fundus image, considering conditions like bleeding or tumorous tissue that may hinder the detection of bifurcations or vascular tree segmentation. They also accounted for the curvature of the retina, treating the fundus image as part of a 3D structure for mapping.

In [22], similarity and local weighted mean transformations were employed, selecting the registration with the best accuracy as the final result. The authors developed a feature extraction method based on detection of local extrema in a hierarchical Gaussian-scale space. The detected feature points detected were located over blood vessels, which were then described using scale-invariant feature transform (SIFT) [23].

Moving towards deep learning approaches, Benvenuto et al. [24] presented an unsupervised deep learning network for deformable fundus image registration. They utilized convolutional neural networks (CNNs) and a spatial transformation network to achieve registration without requiring ground-truth data. Wang et al. [25], on the other hand, utilized deep learning for blood vessel segmentation using a Deep Retinal Image Understanding (DRIU) network, employing a pretrained model as a feature detector and descriptor. Their registration approach utilized a homographic model for warping to map the source image onto the target image.

Feng et al. [26] took a different approach, employing a fully convolutional neural network model to segment vascular structures in retinal images and detect vascular bifurcations. These bifurcations were used as feature points on the vascular mask, and transformation parameters for registration were estimated based on the correspondence of these feature points. They employed homography for registration.

Similarly, Rivas-Villar et al. [20] adopted deep learning for the detection of bifurcations and crossovers of the retinal blood vessels. Their method, which represents the first deep learning feature-based registration method in fundus imaging, utilized a neural network for keypoint detection, and RANSAC (random sample consensus) was used to match the key points, without the need for complex descriptors.

An alternative approach was presented in [27], which stands as one of the prominent methods for evaluating fundus image registration across the FIRE dataset. This technique incorporates both blood vessel bifurcations and the SIFT detector as feature points, initiating an initial camera pose estimation using RANSAC with a spherical eye model, leading to accurate outcomes. Subsequently, parameters for an ellipsoidal eye model are estimated, and the camera pose is further refined. Despite its remarkable accuracy, a notable weakness arises in its feature detection strategy, as it employs two feature detectors simultaneously, yielding a greater number of feature points. This, in turn, elevates the computational complexity of the matching and registration process, thus extending processing times. Moreover, utilizing SIFT as a feature detector in fundus images results in multiple points situated at the image’s edges, contributing feature points that might be deemed noise due to their insignificance in the matching process.

Regardless of whether deep learning is employed for feature detection, it remains crucial to identify features that effectively represent the image’s characteristics. Commonly extracted anatomical features from fundus images include blood vessel bifurcations [20,28], optic discs [29,30,31], and fovea [31,32]. Widely used feature extractors encompass SIFT [23], SURF [33], and the Harris corner detector [34]. While some works suggest that utilizing blood vessel bifurcations as feature points may lead to inaccuracies when disease is present in fundus images [21,26,35], this method adopts these features due to their invariability concerning intensity, scale, and rotation. The proposed method is evaluated using the FIRE dataset [18], a collection of images that includes a category focused on images with anatomical changes in the registrable pairs.

In contrast to the aforementioned works, this research focuses on fundus image registration based on bifurcations detected over the skeleton of segmented blood vessels. The decision not to depend on deep learning for feature extraction or matching is made to secure robustness and independence from large quantities of training data. Additionally, by using a public dataset for evaluation, this method enables direct comparisons with other techniques, facilitating the identification of improvements in fundus image registration. Although there are concerns regarding the reliability of bifurcations as feature points due to possible obstructions from tumorous tissue, the issue is addressed by utilizing a similarity transformation matrix that necessitates only two correspondences. It is expected that, at the very least, the minimum number of correspondences can be found, even in regions with tumorous tissue.

## 3. Proposed Method

In this section, the concept of the proposed method is introduced. The primary focus of this paper is the development of a feature-based fundus image registration method that utilizes bifurcations and crossovers located over the morphological skeleton of blood vessel segmentation as feature points. As shown in Figure 2, the proposed method comprises four main stages: feature extraction, feature matching, computation of the transformation matrix and image warping, and image blending.

### 3.1. Flowchart Description

The flow chart in Figure 2 provides an overview of the proposed method. The input consists of a pair of color fundus images. First, feature points are extracted, focusing on the vascular tree region. A blood vessel segmentation process is applied to the original fundus images, followed by thinning or skeletonization to detect bifurcation patterns in the thinned images. Secondly, each image’s feature points are represented by descriptors to facilitate matching between source and target images, identifying shared points. Thirdly, once the points in the source image are related to the target image, a transformation matrix is computed and applied to warp the source image. Finally, a blending process is applied to ensure a seamless resulting image, avoiding visible seams caused by potential exposure differences in the overlapped regions of the two images. Further details of the proposed method are explained in the subsequent sections.

### 3.2. Feature Extraction

For feature-based registration methods, the initial step involves feature extraction. In this application, feature points are located over the vascular tree. Therefore, the primary objective is to delimit the area to this specific scope by performing blood vessel segmentation. This crucial stage is based on the segmentation process proposed in [36], which entails obtaining a local coarse vessel segmentation image and a curvature image. Subsequently, morphological reconstruction is conducted using the curvature map as a marker and the coarse vessel segmentation image as a mask to generate the final vessel map.

This method emphasizes avoiding segmentation problems caused by central vessel reflex. The presence of central vessel reflex can lead to variations in intensity between the inner part of the vessel and its boundary, resulting in segmentation difficulties and occasionally producing undesired outcomes, such as gaps or hollows. This effect may cause erroneous bifurcations, making the avoidance of this issue crucial for the proposed method. Figure 3 displays the results of the coarse vessel segmentation, demonstrating successful mitigation of this problem.

After the segmentation stage, a thinning process is applied. In this application, the morphological reconstruction from the curvature map is omitted, and only coarse vessel segmentation is utilized to expedite the process. Furthermore, retaining coarse vessel segmentation allows for the detection of a few additional points along the vessels during feature detection. Although these points may not be actual vessel bifurcations or crossovers, they still serve as potential reference points in both images for the alignment process. The decision to consider these points stems from the fact that a feature extractor does not necessarily require relevant positions in the image; it only needs detectable points that can be matched between the images for alignment [37,38]. Moreover, limiting the points to bifurcations and crossovers may significantly reduce the number of candidate points [20], complicating the registration process.

The process of obtaining the local coarse vessel image begins with a preprocessing stage. This stage involves applying Gaussian smoothing, followed by the top-hat operator and contrast enhancement, on the inverted green channel of the original fundus image. The process concludes with gray-level intensity mapping to saturate high- and low-intensity values across the fundus image. For this application, 3% of the high- and low-intensity values are saturated to increase contrast and prevent the loss of bifurcations or crossovers. After the preprocessing phase, segmentation is ready to be performed. The segmentation method primarily relies on the cumulative distribution function of intensities over the image, which reveals a consistent percentage of pixels corresponding to the vascular tree within the fundus image. According to [39,40,41], this percentage ranges from 10.4% to 14.9%. Based on this evidence, first, a global threshold value ($\tau_{i}$) is determined from the cumulative distribution function. This value is obtained by identifying the maximum distance between the identity function and the curve of the cumulative distribution function calculated using Equation (Equation 1): (1)$$\tau_{i}=\underset{l\in \left\{0-255\right\}}{\mathrm{argmax}}\;\left\{\frac{\left|f\left(l\right)+\Delta_{f}\cdot \left(1-l\right)-1\right|}{\sqrt{1+\Delta_{f}^{2}}}\right\},$$
where f(l) represents the cumulative distribution function of intensities (*l*), and $\Delta_{f}$ corresponds to the difference between the cumulative probability of the highest and lowest image intensities.

Subsequently, by using a normalized gray-level co-occurrence matrix (NGLCM) [42] that is omnidirectional and normalized with a distance of d=1, the probability of background pixel intensity in the background region ($P_{bg}$) is calculated. This probability is obtained by summing the probabilities of pixels with intensities of less than $\tau_{i}$ that have surrounding pixels with intensities of less than $\tau_{i}$. The obtained probability, along with the correlation of the NGLCM, is used to calculate the maximum number of pixels in a square window (*W*) of size n×n that are likely to belong to tissue intensities allowed over a vessel for a given pixel in the preprocessed image. This is expressed by Equation (Equation 2): (2)$$\tau_{sd}=\left\lfloor P_{bg}\cdot corr\cdot \#\left\{W\right\}\right\rfloor$$
where $\#\left\{W\right\}$ corresponds to the cardinality of *W*, $\left\lfloor \cdot \right\rfloor$ is the largest integer less than or equal to the operand, and corr stands for the correlation of the NGLCM defined by Equation (Equation 3): (3)$$corr=\sum_{x,y=l_{\min}}^{l_{\max}}P_{x,y}\left(\frac{\left(x-\mu_{x}\right)\left(y-\mu_{y}\right)}{\sigma_{x}\sigma_{y}}\right),$$
where μ and σ represent the mean and standard deviation of the pixel intensities, respectively.

The final segmentation is achieved through a pixel-by-pixel evaluation of the preprocessed image. A given pixel (*p*) centered in window *W* is classified as part of the background if the number of pixels in *W* with background intensities exceeds $\tau_{sd}$. Otherwise, pixel *p* is considered part of the vascular tree. An example of the coarse vessel segmentation results used in this proposal is shown in Figure 4.

Bifurcations and crossovers over the vascular tree exhibit a vessel’s characteristic of dividing itself in two, presenting a T-shaped pattern in different orientations. However, the vessel thickness varies between these patterns, so they are generalized by reducing the whole vascular tree’s thickness to one pixel. To achieve this, a skeletonization method is required. The method proposed by Zhang and Suen in [43] has proven to be effective in preserving the foreground connection while thinning the vascular tree to one pixel.

This thinning method analyzes all pixels in the binary image and their direct neighbors. The analyzed pixel features include the number of non-zero pixels in the eight surrounding pixels; the number of zero-to-one transitions in the surrounding pixels in clockwise order; and the product of the four possible triads of pixels combined from the up, down, left, and right pixels. These features are analyzed in two separate steps, which are iterated. In each iteration, pixels are modified to zero based on the results of these two steps. The process continues until no further changes occur in the image. The result is a binary skeleton with one-pixel thickness, preserving the original connectivity. Figure 5 displays an example of the resulting skeletonization stage, in which (**a**) highlights bifurcations found over the vessels; these bifurcations are also highlighted over the skeleton to confirm the effectiveness of preserving the connectivity present in the vessel segmentation.

Feature points are located by examining patterns present over the skeleton. Different orientations of T-shaped patterns are defined to identify these particular patterns of foreground and background pixels found in the vessel segmentation skeleton. Hit-or-Miss transform, a binary morphological operation using two structuring elements ($B_{1}$ and $B_{2}$), is employed to represent the foreground (i.e., the morphological skeleton of blood vessel segmentation) and the background of the searched patterns. The bifurcation patterns over the skeleton image (*A*) are defined by Equation (Equation 4): (4)$$A⊛B=\left(A⊖B_{1}\right)\cap \left(A^{c}⊖B_{2}\right)$$
where ⊛, ⊖, and ∩ represent convolution, erosion, and intersection operators, respectively; *B* is a combination of structuring elements, where $B_{1}$ represents the pattern’s skeleton, and $B_{2}$ represents the pattern’s background; and $A^{c}$ denotes the complement of the skeleton image.

Bifurcations exhibit various patterns after the skeletonization process. Figure 6a shows the skeleton image, with framed areas enlarged in Figure 6b to highlight the patterns present at the bifurcation locations. It is evident that multiple patterns are present. Therefore, in Figure 6c, the structuring elements (*B*) used to detect the bifurcations are presented.

After detecting these patterns, the positions of bifurcations are identified, defining the feature points. Figure 7 illustrates the result of feature extraction from a fundus image.

### 3.3. Feature Matching

The association of the source and target images requires determination of which features are present in both images. Since fundus images lack relevant texture and the background encompasses the majority of the fundus image, a robust feature descriptor is needed. In this approach, a histogram of oriented gradients (HOG) is utilized as a local descriptor, capturing gradient information from the surrounding region of a given position and generating a feature vector based on gradient magnitude and orientation.

Before calculating the HOG descriptor for each image region, normalization is applied to reduce the impact of illumination effects. These effects may occur in registrable fundus image pairs taken from different angles, by different devices, or at different times. Each region is divided into blocks, and the gradient magnitudes of each block are distributed in a histogram according to their direction. The final descriptor is obtained by concatenating these histograms.

In this application, considering the favorable performance of the SIFT descriptor in several applications and its variants [22,44,45,46,47,48,49], the analyzed areas and blocks are replicated. For each feature point, a 16 × 16 pixel area is taken and divided into blocks of 4 × 4 pixels. This descriptor uses an eight-direction orientation histogram, resulting in a 128-dimensional feature vector for each point. Its main advantage is its invariance in response to geometric and photometric transformations.

After describing each feature point for both the source and target image, it is possible to compare feature points from the two images to find correspondences.

In ideal circumstances, each feature point and its descriptor on the source image should be distinct enough to be associated with their respective most similar feature point in the target image (i.e., nearest-neighbor matching strategy). However, the general lack of texture in fundus images, due to the predominance of the background, necessitates employing a stricter criterion when relating features. Therefore, the second-nearest-neighbor matching strategy is utilized to retain sufficiently distinct matches. The threshold for determining a valid match is set as a fraction of the magnitude of the most similar vector; for this proposal, a threshold of 0.2 is experimentally set.

At this point, candidate matches found between fundus images are determined. To reduce the number of outliers, a cross check is applied, meaning that the matching process is repeated in reverse, finding matches from the source image to the target image and vice versa. Figure 8 shows the result of these two sets of matches and the result of the cross checking, which consists of the matches found in both directions. Despite the clear decrease in the number of matches, the result is an increased inlier ratio, which facilitates the computation of the following transformation model.

### 3.4. Transformation Matrix Computation and Image Warping

Following feature matching between fundus images, the correspondences are employed to map the source image to the target image. A 2D planar transformation model is used to represent this relation. Given a point (*P*) over a planar surface, with projections ($p_{1}$ and $p_{2}$) over two images, the transformation matrix (*T*) represents the transformation that relates $p_{1}$ and $p_{2}$ for any given point on the surface; this means $p_{2}=Tp_{1}$. Different planar transformation models may be employed to relate the images, each preserving different features. Projective, affine, and similarity transformations have been used to register fundus image pairs in different applications [20,22,26,37,50,51,52,53]. Although some fundus registration applications employ these or more complex transformation models, in [54], it was demonstrated that the optimal mapping differs depending on the registered image pair. Then, since similarity and affine transformations have the advantage of requiring fewer correspondences for computation of their matrix and the simplicity of their computation, such models employed in this application, as some image pairs with anatomical changes caused by diseases may have a scarce number of correspondences.

The validation of this method with multiple models is motivated by the presence of so-called “localization error” and the varying degrees of freedom for each model. Localization error refers to the discrepancy between the detected location of a feature point and its actual or expected position (e.g., the difference between the detected bifurcation position and its exact location) in fundus images. Consequently, the localization error for a given bifurcation landmark in the source and target images may differ in distance and orientation. Considering this error is crucial when selecting a transformation model, as different models allow for varying degrees of deformation during mapping, resulting in distinct impacts of the localization error for each model. This localization error leads to a registration error, where the exact locations of points in the overlapped region do not coincide precisely when mapping the source image to the target image. This effect causes a “double exposure” artifact in the final result. Figure 9 shows a representation of these two types of errors involved in the registration process.

However, localization error is not the sole source of registration error; the selected transformation model also plays a significant role, as it represents a 3D curved surface on a 2D plane. Therefore, it is essential to consider a different model that accounts for this factor, such as those proposed in [21,27,55]. However, this work is limited to employing the similarity and affine planar transformation models because of their previously specified benefits.

The transformation matrices are computed using the Random sample consensus (RANSAC) algorithm, a resampling method that constructs models based on the minimum required observations and that widely used in image registration to eliminate incorrect matches [56,57]. It selects the final model based on the number of inliers. For each transformation model, a different number of matches is needed to compute the corresponding matrix. Affine transformation requires a minimum of three correspondences, while similarity only needs two, streamlining the fundus registration process when few correspondences are detected.

When analyzing the feature points in the fundus image, noise may cause some of these points to be in close proximity to each other. As a result, certain points could be mistakenly selected during computation of the transformation matrix by RANSAC. To address this, a constraint is introduced in this proposal, ensuring a minimum distance between randomly selected matches. This constraint prevents the inclusion of matches originating from nearby feature points. Moreover, by selecting matches that are well distributed, the impact of localization error is reduced, leading to a decrease in registration error. Equation (Equation 5) defines the number of iterations (*N*) required to find a model: (5)$$N=\frac{log\left(1-p\right)}{log\left(1-{\left(1-v\right)}^{m}\right)}$$
where *p* represents the probability that at least one of the randomly selected sets does not include an outlier (commonly set to 0.99), *v* signifies the probability of any selected data point being an outlier, and *m* denotes the minimum number of observations required to construct the model.

### 3.5. Image Blending

The image-warping process is initiated to create a single, wider fundus image based on the geometric relation between the two fundus images. While ideal conditions would entail uniform exposure in the registered images, practical scenarios deviate from this expectation due to non-uniform illumination during fundus image acquisition. Consequently, exposure differences emerge in the overlapped regions of the source and target images. Shading artifacts originating from internal reflections further contribute to exposure variations in certain regions [58]. Attempting to rectify this by simply replacing the intensity values of the overlapped area with those from the source image is unsatisfactory, as it would create an undesirable visual effect resembling a noticeable seam. Consequently, a more advanced technique is required to seamlessly join the two images, ensuring that no visible edges or seams are apparent. The multiresolution image blending technique proposed in [59] emerges as a highly effective alternative to address this issue.

The employed method is initiated by constructing Laplacian pyramids for the two images. These pyramids are generated through image subsampling, followed by upsampling and the computation of the difference between each level and its previous octave, yielding the respective Laplacian for the octave. Next, element-wise multiplication occurs between the Laplacian and the fundus image’s mask, after which the Laplacians from both images are added. Since there is an overlapping region, preserving exposure and preventing overflow necessitate division of the Laplacian addition by the sum of the masks from both the source and target images, resulting in the fused Laplacians. Finally, the blending result is obtained by upsampling the fused Laplacian from the last octave and adding it to the fused Laplacian from the previous octave, with this process repeated until the original dimension is restored. Figure 10 depicts an illustration of this image-blending process.

## 4. Experimental Results

This section evaluates the efficacy of the proposed method using a public database. First, the selection of the database for evaluation is introduced and justified. Next, the evaluation metrics are defined, and a quantitative analysis comparing the proposed method with related works is performed. Finally, some examples of successful results obtained with the proposed method are presented.

### 4.1. Dataset

This study exclusively uses the Fundus Image Registration (FIRE) dataset for evaluation. The choice of this dataset is driven by the scarcity of public databases specifically designed for fundus image registration. To date, there are only four databases with registrable image pairs (e-ophtha [60], RODREP [61], VARIA [62], and FIRE [63]). Among these datasets, FIRE stands out as the only one providing a ground truth for registration with ten control points. This feature allows for quantitative evaluation of the method’s performance and facilitates proper comparisons with previous works.

The FIRE dataset comprises 129 retinal images with a resolution of 2912 × 2912 and a field of view of 45°, forming 134 image pairs [18]. These images belong to 39 patients ranging in age from 19 to 67 years and are categorized into three different classes, each with a distinct registration application: category S is intended for super resolution, category P for mosaicking, and category A for longitudinal studies. Among these categories, only category A contains registrable pairs with anatomical changes, such as vessel tortuosity, microaneurysms, and cotton-wool spots. Table 1 provides a summary of the characteristics of the FIRE dataset, while Figure 11 illustrates the registrable image pairs, along with their respective ground-truth control points, for each category.

### 4.2. Evaluation Metrics

The accuracy of the proposed method was assessed using the approach described in [18]. Expertly marked control points strategically distributed on the overlap of each fundus image pair were used for registration error measurement. The registration error is computed as the mean distance between each control point in the target image and its corresponding point in the source image after registration. This evaluation was conducted for the entire dataset, as well as for each category separately.

To visualize the accuracy across different error thresholds, a 2D plot was generated. The x axis represents the error thresholds, while the y axis shows the percentage of successfully registered image pairs for each threshold. A successful registration occurs when the error is below the specified threshold. The resulting curve illustrates the success rate as a function of target accuracy, enabling a comparison of different methods and facilitating the selection of the most suitable one based on the accuracy requirements of specific applications.

This error analysis methodology is particularly beneficial when introducing image registration methods into clinical practice, as it contributes to determining the necessary level of accuracy required for clinical purposes [64,65]. Additionally, the curve is used to provide a comprehensive evaluation of the method through the analysis of the area under the curve (AUC).

### 4.3. Performance on the FIRE Public Dataset

Figure 12 illustrates the performance of diverse registration methods on FIRE dataset, analyzing accuracy for each category separately and overall. Additionally, Table 2 presents the area under the curve (AUC) for all compared methods. The method’s registration error in comparison with two basic models, namely the similarity and affine transformation models, is also analyzed.

The method proposed by Yang et al. in [37] aims to register natural and man-made scenes, as well as medical images, including retinal images. Their method uses corners and face points as feature points. Depending on the image pair being aligned, they employ different transformation models, such as similarity, affine, homography, and quadratic models.

Similarly, Chen et al. [50] used corners identified by a Harris detector as their feature points. Based on the number of matches obtained after feature extraction, description, and matching, different models are used. Like the proposed method, their descriptor is also gradient-based, considering the main orientation of the points.

It is evident from Figure 12 that results vary significantly across different categories. Categories S and A, with larger overlapping regions, are expected to yield a higher number of correspondences, leading to better matching performance. Conversely, category P may face challenges, as the number of potential matches is limited to the overlapping area between fundus images. In category S, the GDB-ICP method shows outstanding performance, with numerous successful registrations and small registration errors. However, it achieves successful registration for only 84.5% of the category, while both the proposed method and the Harris-PIIFD method achieve 100% success, with a greater tolerance of registration errors. In category A, the Harris-PIIFD method stands out, with a success rate of 78.51%, while the proposed method reaches a maximum of 57.15%. Each method in this category shows low success rates for small registration errors. Furthermore, for category P, the GDB-ICP method outperforms the other presented methods, but with a greater tolerance to error, the proposed method’s success rate surpasses that of GDB-ICP. Table 3 provides more details on the registration errors for the methods compared in Figure 12, which are similar that of the method presented in this paper in terms of feature extraction. By analyzing the area under the curve (AUC) (Table 2), it is evident that the proposed method outperforms GDB-ICP in both category S and category A. In category P, the proposed method’s performance surpasses that of the Harris-PIIFD method.

However, the REMPE method (H-M 17) [27] surpasses the performance of that of all the previously mentioned techniques, achieving an outstanding area under the curve (AUC) of 0.773 for the complete dataset, thereby emerging as the top-performing approach within this dataset. The study in which the REMPE method primarily was proposed primarily centered on estimating eye pose and shape in order to register fundus images, employing the extracted feature points from bifurcations and SIFT as a basis for 3D pose estimation and utilizing various eye models for improved accuracy.

Table 4 provides a comprehensive comparison of the features associated with the methods depicted in Figure 12. Notably, the method proposed in this work exhibits similarities to the Harris-PIIFD and GDB-ICP approaches. This observation sets the stage for a more detailed comparison of these methods, as elaborated upon in Table 3. Table 3 shows that the proposed method surpasses other presented methods in terms of the number of successful registrations. These methods correspond to the top methods that are similar to our approach in terms of the registration process. However, it is also true that our method exhibits slightly less precision compared to the GDB-ICP method, which stands out for its high precision but with a lower number of successfully registered image pairs. Moreover, our proposal demonstrates increased registration error and standard deviation when compared to the other methods. These errors can be attributed to two factors: localization error and the selection of matches during the RANSAC process. During feature extraction, some inaccuracies from segmentation persist, leading to discrepancies between points in the source image and their corresponding points in the target image. Nonetheless, due to the similarity of the regions and the minor localization error, the descriptors are deemed sufficiently similar and are included in the RANSAC process. Figure 13 showcases sample results obtained for each category of the FIRE dataset, illustrating correct matches and the results after the registration process.

## 5. Discussion

The primary objective of this research was to develop a feature-based fundus image registration method utilizing blood vessel bifurcations as feature points. The aim was to contribute to a baseline for fundus image registration and address the absence of validation for previously proposed methods in the field. A comparison with other registration methods was conducted, and the performance of the technique was evaluated using registration error as the key metric.

The proposed method involves a multistep process to achieve fundus image registration. The registration process is initiated by employing a robust blood vessel segmentation technique to define the region of interest. The segmentation process plays a crucial role in accurately detecting feature points within the fundus image. A thinning or skeletonization process is applied to the segmented blood vessels to achieve single-pixel thickness, enabling the detection of bifurcation patterns over the vessel skeleton. The detected bifurcation patterns are used as feature points for the registration process. These feature points are represented using a histogram of oriented gradients (HOG) descriptors. The correspondences between feature points are established using the second-nearest-neighbor and random sample consensus (RANSAC) algorithms to remove outliers. Image registration is performed using similarity transformation and affine transformation models, with similarity transformation demonstrating superior accuracy for this application. Lastly, image blending based on Laplacian pyramid is applied to create a seamless registered fundus image.

Comparable performance to that of other similar techniques was exhibited by our method, which, despite its relative simplicity, displayed strength in the distribution of feature points over the entire fundus image. Unlike more general-purpose feature detectors like SIFT, SURF, or ORB, this approach ensures the distribution of feature points throughout the image, potentially leading to more accurate registration results. Moreover, these general-purpose feature detectors tend to detect numerous features over the fundus image outline, as demonstrated in [26], generating undesired outliers and slowing the feature-matching process.

Several challenges were encountered during the research despite the promising results. The development of a robust method for blood vessel segmentation proved to be the most significant challenge, as the accuracy of feature point detection heavily relies on the quality of the segmentation process. For feature points, a histogram of oriented gradients (HOG) descriptor was used, which outperformed other descriptors such as local binary patterns (LBPs) and fast retina keypoint (FREAK). The potential for further improvement in overall registration accuracy is recognized by considering the adoption of even more robust segmentation techniques, such as convolutional neural-network-based methods.

Another challenge emerged in obtaining a suitable dataset for quantitative evaluation. Due to the lack of annotated data specifically designed for fundus image registration, the FIRE dataset was utilized, enabling a significant quantitative evaluation. The scarcity of annotated data not only hinders the comparison of techniques but also imposes limitations on the development of deep-learning-based approaches that require large amounts of data.

The proposed feature-based fundus image registration method holds potential applications in the medical field. One of the main uses is assisting in the evaluation of retinal diseases. By enabling a broader field of view and aligning fundus images for comparison in longitudinal studies, this method can aid in the early detection and monitoring of retinal pathologies. However, the level of accuracy required for fundus image registration to be incorporated into clinical practice remains uncertain, which may influence its practical implementation.

## 6. Conclusions and Future Work

This research presents a viable approach for feature-based fundus image registration using blood vessel bifurcations as feature points. The method demonstrates competitive performance compared to existing techniques, showcasing the reliability of bifurcations in registering fundus images. The significance of achieving feature point distribution lies in the potential to improve registration accuracy by reducing localization errors. While a dispersed distribution is exhibited by the method, it is acknowledged that further enhancements can be achieved by improving blood vessel segmentation and feature point detection.

Challenges related to feature-based registration using bifurcations are addressed by our research, accommodating scenarios where blood vessels may be obstructed by retinal pathologies or tumorous tissues. Potential feature points can be detected by the method even in regions with minimal bifurcations, rendering it robust and adaptable. Practical applications encompass assisting ophthalmologists in retinal analysis for longitudinal studies, treatment comparisons, and disease progression monitoring.

For future work, the method will be optimized for real-time applications by exploring efficient algorithms for feature detection and image blending. Stronger feature descriptors will be investigated to improve the matching process while balancing computational efficiency and accuracy. Additionally, the need for larger annotated datasets is emphasized to facilitate the comparison of future methods and the development of deep-learning-based approaches.

In conclusion, this research contributes to the advancement of fundus image registration by providing an effective and reliable method based on blood vessel bifurcations. As optimizations and further advancements are explored, the proposed approach holds promise for practical applications in retinal analysis and medical imaging domains.

## Figures and Tables

**Figure 1 sensors-23-07809-f001:**
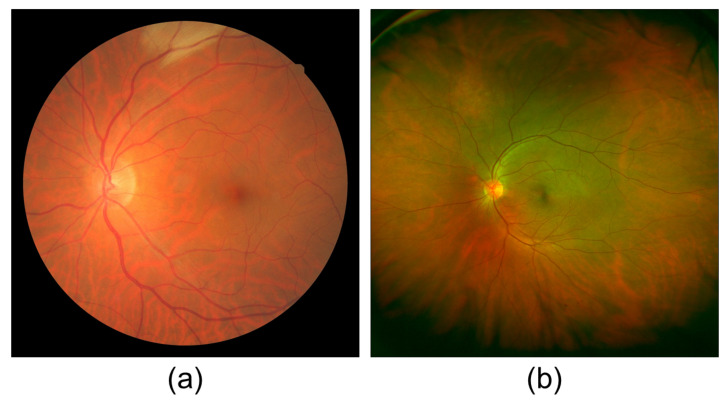
Comparison of the field of view between (**a**) a conventional fundus image [18] and (**b**) a wide-angle fundus image. The images correspond to different patients and are used solely to contrast the scope difference between the two techniques.

**Figure 2 sensors-23-07809-f002:**
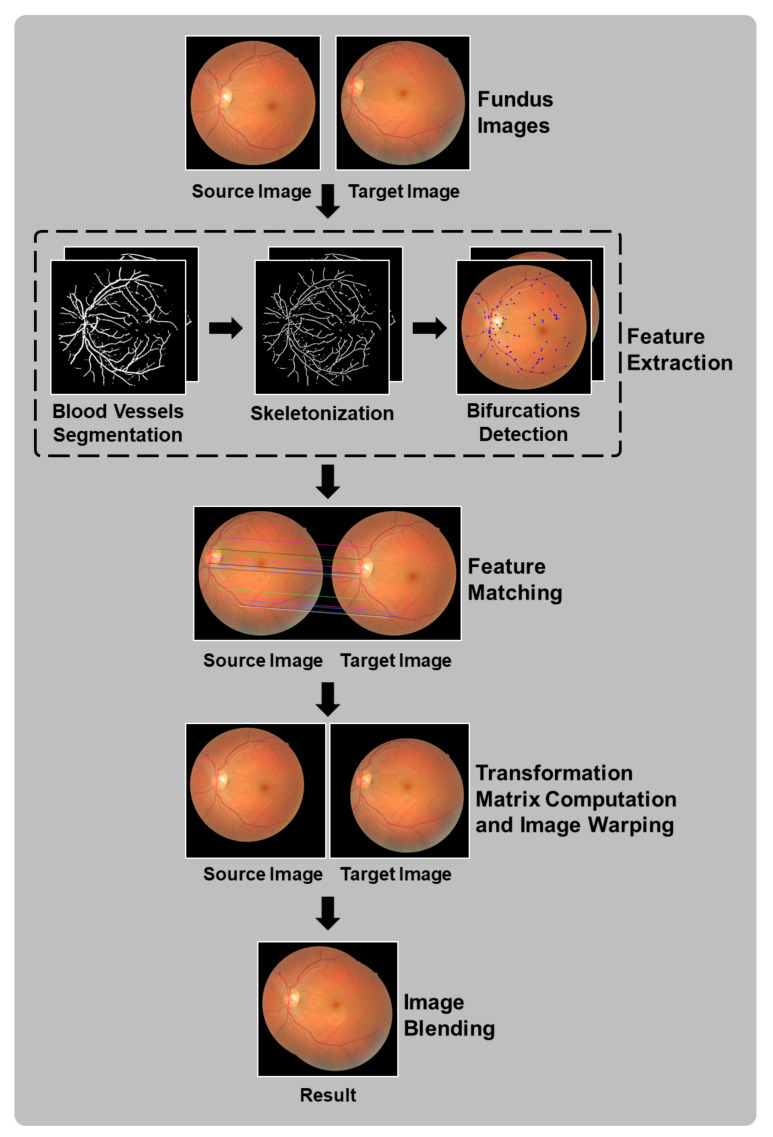
Flow chart of the proposed method, with stages involving processing of both source and target images shown as steps with two images.

**Figure 3 sensors-23-07809-f003:**
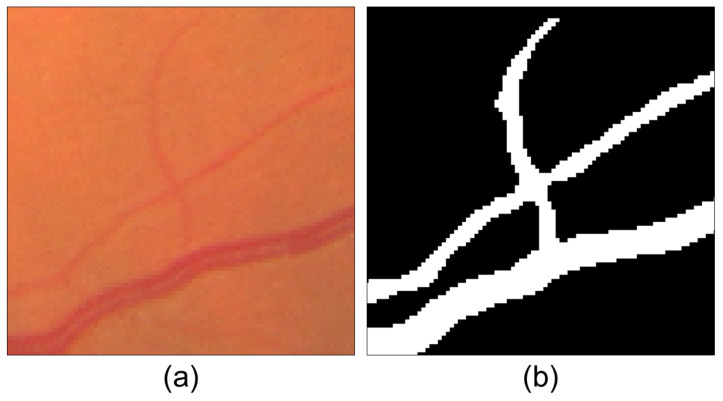
Comparison of (**a**) a section of a fundus image with vessel reflex over a major vessel and (**b**) the result of coarse vessel segmentation, where the problem of vessel reflex is successfully avoided.

**Figure 4 sensors-23-07809-f004:**
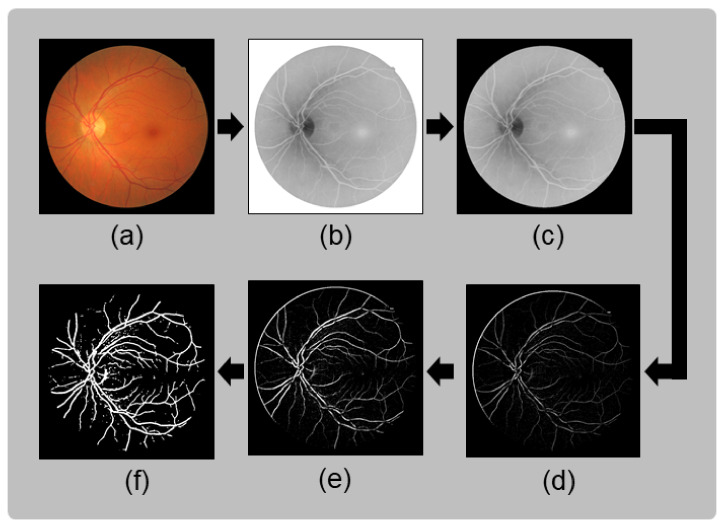
Stages of coarse vessel segmentation: (**a**) original fundus image; (**b**) green channel inverted; (**c**) applying mask and Gaussian smoothing over the green channel; (**d**) top-hat transform applied over the smoothed image; (**e**) histogram stretching over the top-hat transform to increase the contrast between vessels and background; (**f**) coarse vessel segmentation result.

**Figure 5 sensors-23-07809-f005:**
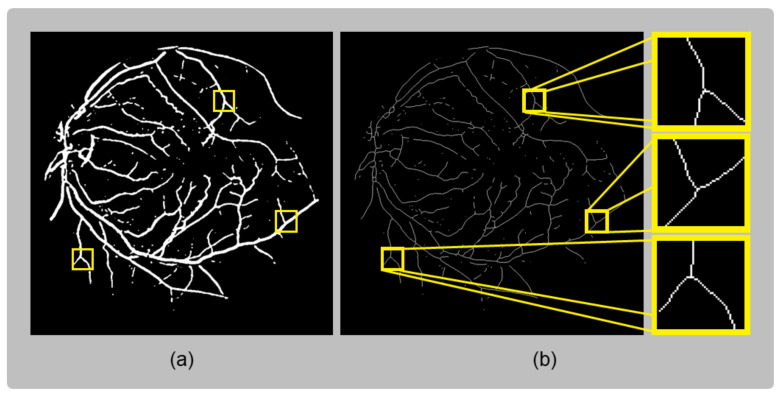
Comparison of (**a**) The original coarse vessel segmentation and (**b**) the skeleton obtained using the Zhang–Suen algorithm. The framed areas in (**a**,**b**) correspond to regions with bifurcations, which are highlighted to demonstrate the efficiency of the skeletonization method in preserving connections.

**Figure 6 sensors-23-07809-f006:**
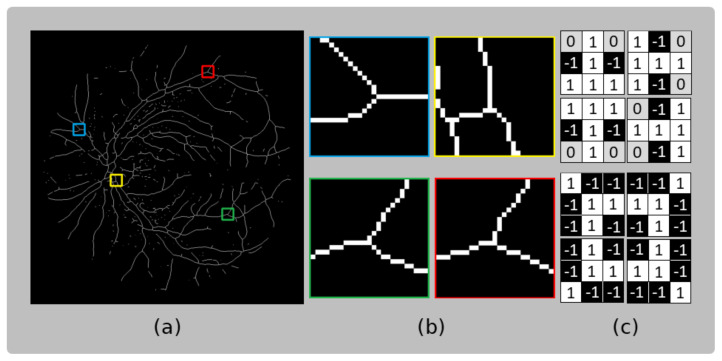
(**a**) Skeleton of fundus image (*A*) and (**b**) samples of bifurcation patterns detected over the skeleton. (**c**) Structuring elements (*B*), where white, black, and gray pixels represent the skeleton pattern, background, and the indifference of the searched pattern, respectively.

**Figure 7 sensors-23-07809-f007:**
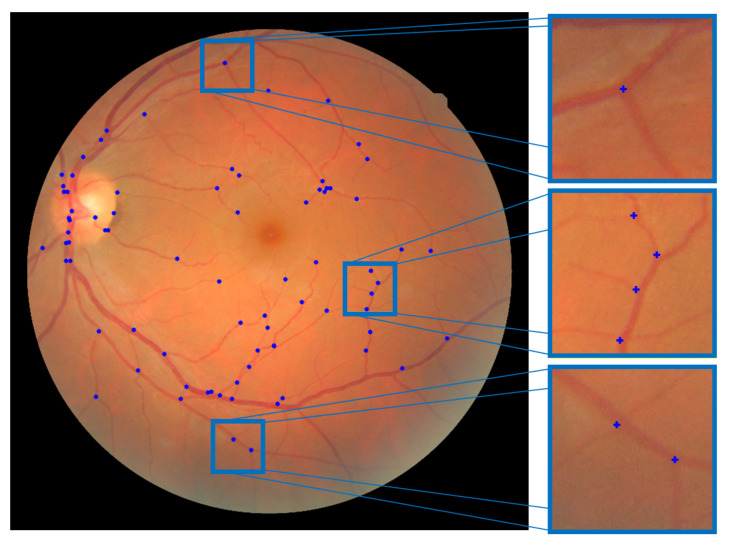
Fundus image displaying the extracted feature points used in the fundus registration process.

**Figure 8 sensors-23-07809-f008:**
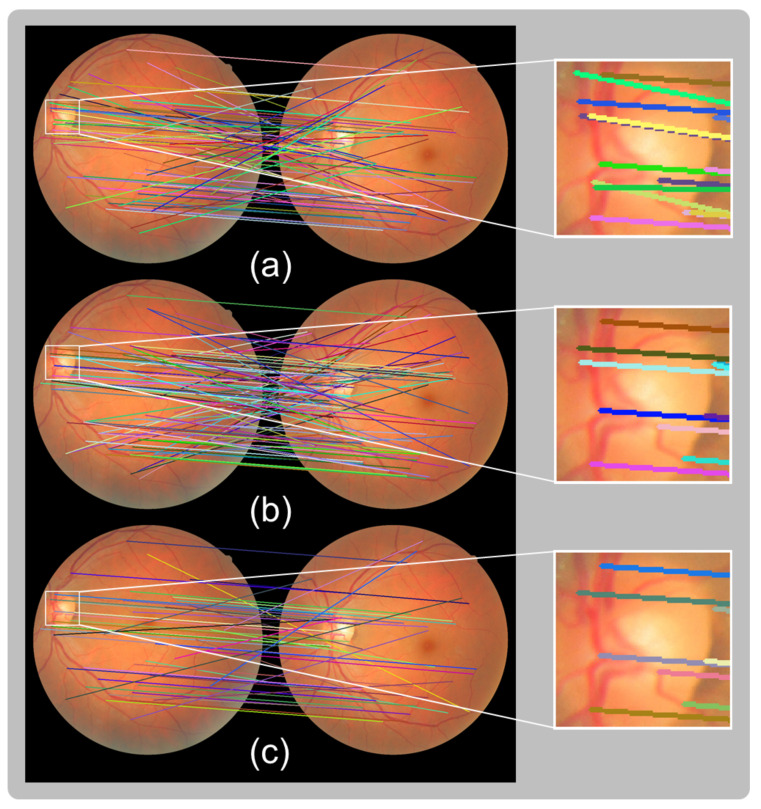
Matches obtained using second-nearest-neighbor matching from (**a**) source to target image and (**b**) from target to source image and (**c**) the result of cross checking. After cross checking, the density of inliers is notably increased. The white zoomed-in boxes show that (**a**) various match lines point in different directions, although (**c**) these matches are dismissed, leading to an increased number of inlier matches. Matches are marked with different colors to enhance the visualization.

**Figure 9 sensors-23-07809-f009:**
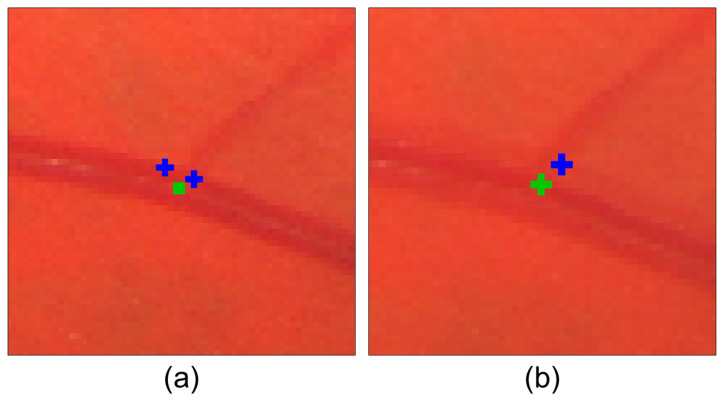
Representation of (**a**) localization error and (**b**) registration error. In (**a**), the green square represents the true position of the bifurcation, while the blue crosses indicate the detected positions for the same bifurcation in the source and target images. The distance between the true position and each detected position is known as localization error. In (**b**), the green and blue crosses illustrate the same locations on the fundus images. However, after aligning the images, the positions differ, resulting in registration error, which leads to a loss of sharpness.

**Figure 10 sensors-23-07809-f010:**
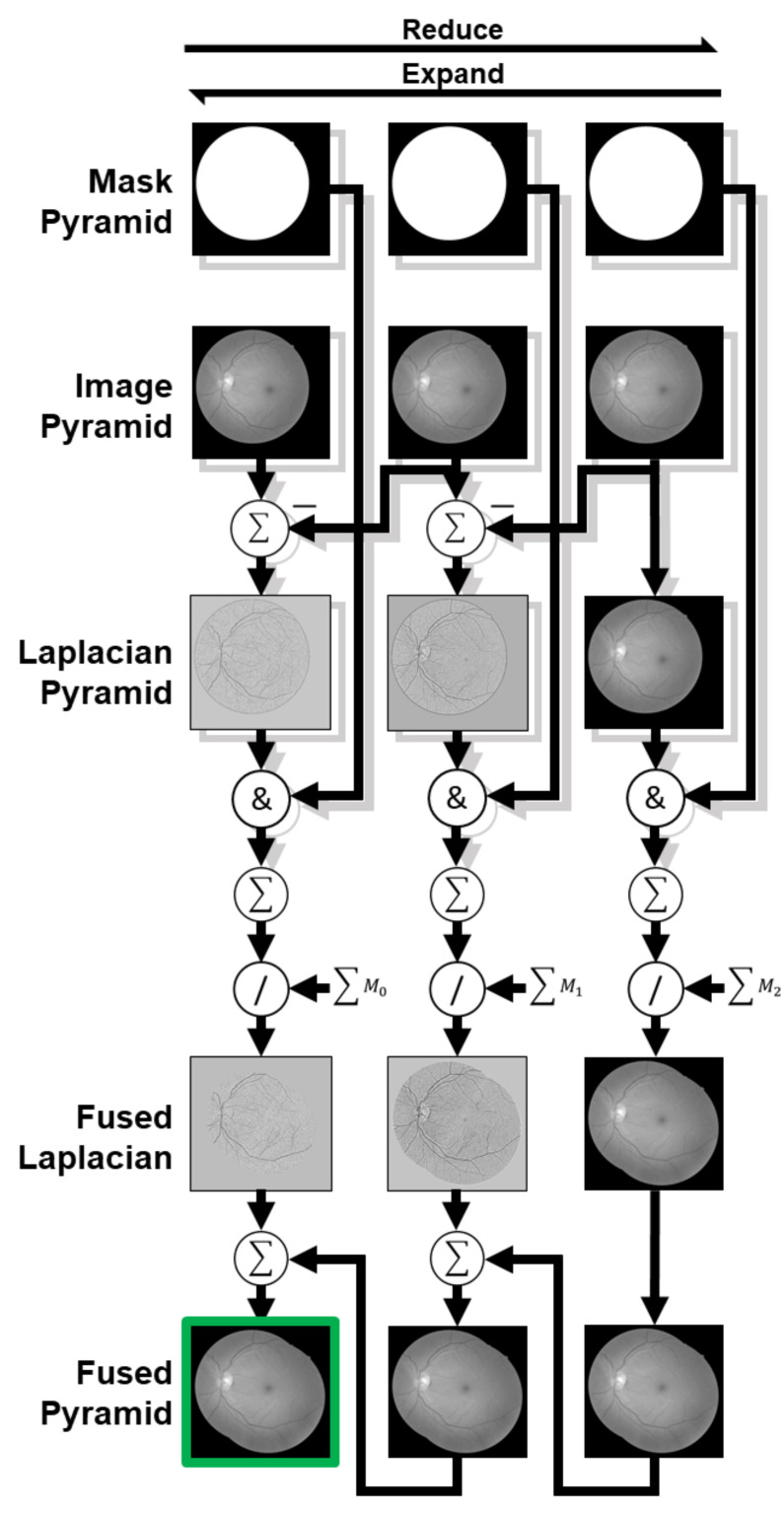
Laplacian pyramid image blending process over three octaves. $\sum M_{n}$ represents the sum of masks from both source and target images in the *n*-th octave. Light-gray shaded stages represent the processing of source and target images. Laplacian pyramid and fused Laplacian images are visually enhanced for improved contrast and brightness.

**Figure 11 sensors-23-07809-f011:**
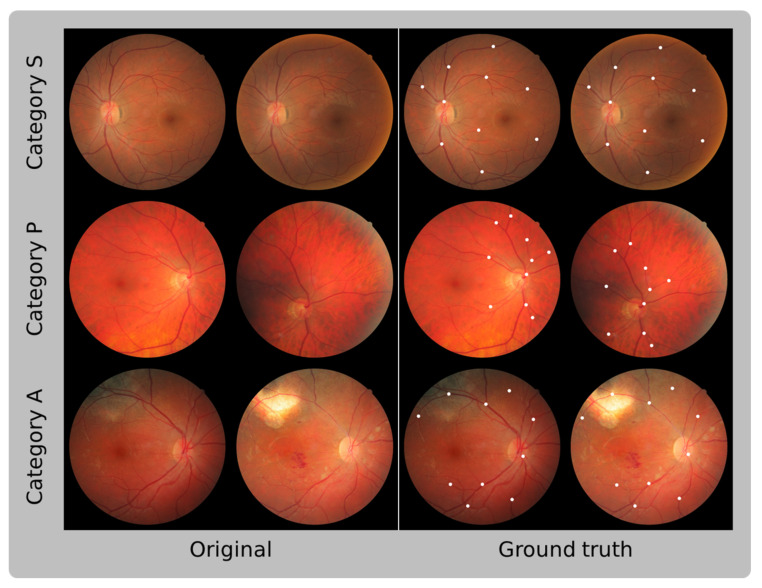
Registrable fundus image pairs from the three different categories of the FIRE dataset. The left column exhibits the original fundus images, while the right column displays the same fundus images with their corresponding annotated control points. These control points were are based on the coordinates provided by the dataset as ground truth. The left original images correspond to the left images in the ground truth, and the right original images correspond to the right images in the ground truth.

**Figure 12 sensors-23-07809-f012:**
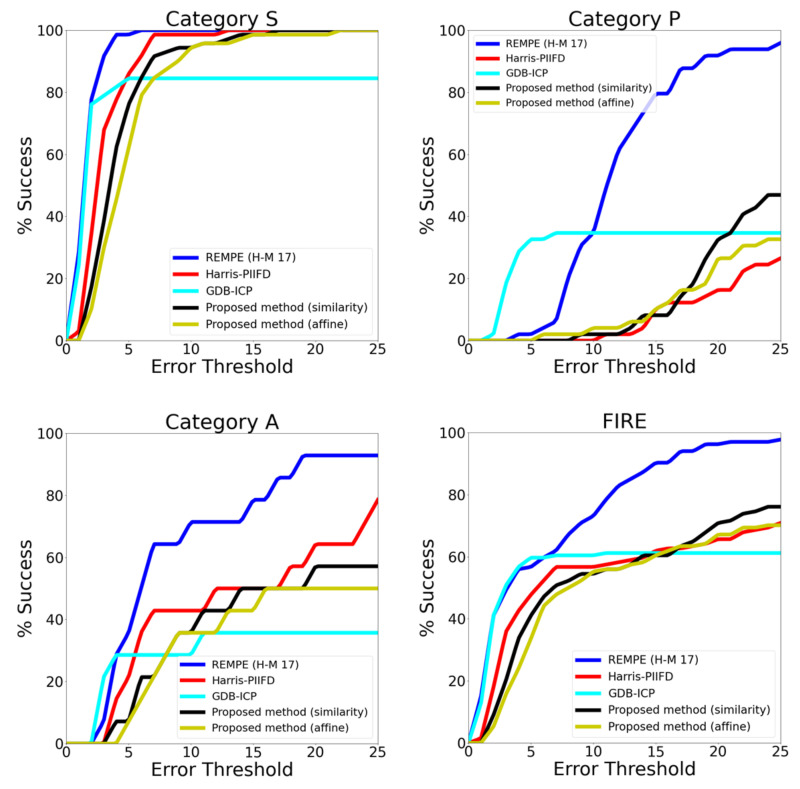
Comparison of retinal image registration methods on the FIRE dataset, including results of the proposed method.

**Figure 13 sensors-23-07809-f013:**
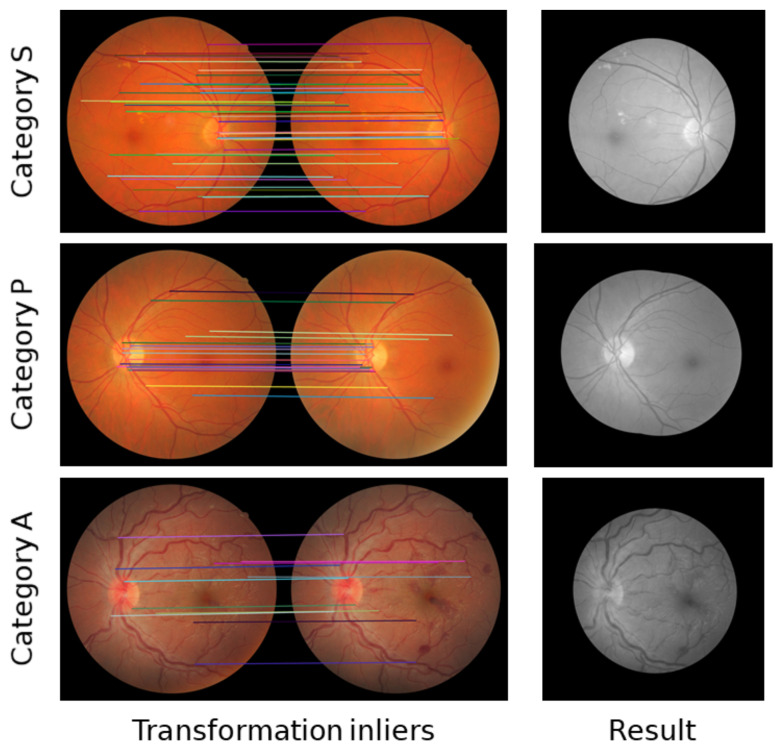
Image registration results of sample image pairs from the FIRE dataset for super-resolution imaging (top row), image mosaicking (central row), and longitudinal study (bottom row).

**Table 1 sensors-23-07809-t001:** FIRE dataset description.

	Category S	Category P	Category A
Total image pairs	71	49	14
Approximate overlap	>75%	<75%	>75%
Anatomical changes	No	No	Yes

**Table 2 sensors-23-07809-t002:** Comparison of area under the curve (AUC) for various retinal image registration methods.

	Category S	Category P	Category A	FIRE
REMPE (H-M 17)	0.958	0.542	0.660	0.773
Harris-PIIFD	0.900	0.090	0.443	0.553
GDB-ICP	0.814	0.303	0.303	0.576
ED-DB-ICP	0.604	0.441	0.497	0.553
SURF + WGTM	0.835	0.061	0.069	0.472
RIR-BS	0.772	0.049	0.124	0.440
EyeSLAM	0.308	0.224	0.269	0.273
ATS-RGM	0.369	0.000	0.147	0.211
Our method ^1^	0.835	0.127	0.360	0.526
Our method ^2^	0.803	0.108	0.328	0.499

^1^ Similarity transformation. ^2^ Affine transformation.

**Table 3 sensors-23-07809-t003:** Comparison of registration error for different retinal image registration methods.

Method	Successful Registrations ^1^	Registration Error (Pixels)			
		**Min**	**Max**	**Mean**	**Standard Deviation**
	Category S				
Harris-PIIFD	71	0.785	12.850	2.981	1.969
GDB-ICP	60	0.486	4.575	1.426	0.777
Proposed method ^2^	71	1.027	16.257	4.114	2.813
Proposed method ^3^	71	1.538	21.425	4.953	3.306
	Category P				
Harris-PIIFD	13	10.041	3870.632	134.862	580.485
GDB-ICP	17	1.946	6.323	3.259	1.133
Proposed method ^2^	23	8.464	1072.128	74.586	177.689
Proposed method ^3^	16	5.180	4457.581	365.182	899.763
	Category A				
Harris-PIIFD	11	3.319	1486.255	149.331	396.753
GDB-ICP	5	2.354	10.416	4.316	3.443
Proposed method ^2^	8	3.300	1302.518	284.825	486.440
Proposed method ^3^	7	4.511	8676.283	1034.973	2316.048
	FIRE				
Harris-PIIFD	95	0.785	3870.632	64.298	367.154
GDB-ICP	82	0.486	10.416	1.990	1.486
Proposed method ^2^	102	1.027	1302.518	59.212	203.945
Proposed method ^3^	94	1.538	8676.283	244.292	958.260

^1^ Registrations with an error of 0–25 pixels. ^2^ Similarity transformation. ^3^ Affine transformation.

**Table 4 sensors-23-07809-t004:** Comparison of method features for fundus image registration on the FIRE dataset, as depicted in Figure 12.

Method	Transformation Model	Strategy	Features Points
REMPE (H-M 17)	Ellipsoid eye model	Camera pose estimation	SIFT and bifurcations
Harris-PIIFD	Polynomial	Transformation model estimation	Corners
GDB-ICP	Quadratic	Transformation model estimation	Corners and edges
Proposed method	Similarity/affine	Transformation model estimation	Bifurcations

## Data Availability

The FIRE database in this study is openly available at https://projects.ics.forth.gr/cvrl/fire/ accessed on 7 September 2023.

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
