# Peer review of "A Straightforward Bifurcation Pattern-Based Fundus Image Registration Method"

_sensors, 2023, doi:10.3390/s23187809_

Round 1

Reviewer 1 Report

This paper presents a straightforward method for registering fundus images based on bifurcations, as these features are the perceptually most significant landmarks distributed over the image, and it aims at contributing to a baseline for fundus image registration based on these ladnmarks as feature points and evaluated over the Fundus Image Registration (FIRE) dataset , the only one publicly available dataset for making a quantitative assessment of registration

error.The topic is interesting but some comments need to be addressed before accepting the manuscript. For example, the abstract is too long and confusing. Also, related work section miss some important papers.

The abstract: The abstract is too long.

 The novelty of the paper is vague. could you please highlight the novelty of the paper?

Introduction

Please do not use long paragraphs.

The introduction section is too long. I advise the authors to split it into related work and an introduction step and remove basic information.

Please highlight the novelty compared to previous work that employed DL for DR diagnosis.

Could you please add the following articles for feature extraction from fundus images using deep learning approaches

GabROP: Gabor Wavelets-Based CAD for Retinopathy of Prematurity Diagnosis via Convolutional Neural Networks

DIAROP: Automated Deep Learning-Based Diagnostic Tool for Retinopathy of Prematurity

Detection and classification of red lesions from retinal images for diabetic retinopathy detection using deep learning models

A Lightweight Robust Deep Learning Model Gained High Accuracy in Classifying a Wide Range of Diabetic Retinopathy Images

An active deep learning method for diabetic retinopathy detection in segmented fundus images using artificial bee colony algorithm

A novel approach for diabetic retinopathy screening using asymmetric deep learning features

Methodology:

English needs revision. Some grammatical are found.

Experimental Results

Please dedicate a section defining performance measures

Please add samples of the dataset and add more details.

Please dedicate a discussion section.

some grammatical mistakes detected.

Reviewer 2 Report

The article is devoted to the problem of bifurcation fundus image for diagnosing eye diseases.

Paper has practical value.

Suggestions:

1. In the introduction, it is necessary to state the main results of the article.

2. The article does not state the research problem (verbal or mathematical).

3. It is necessary to estimate the computational complexity of the developed method.

4. A lot of references are outdated. Please fix it using 3-5 year old papers in high-impact journals.

5. The conclusions in the article are not written specifically. The author must cite the scientific novelty and practical significance of the work.

6. The authors need to improve the quality of figure 12.

Reviewer 3 Report

The manuscript presents a proposal for image registration in the case of a fundus image. The tests were carried out for the FIRE (Fundus Image Registration) dataset.

At this point, the biggest weakness of the reviewed manuscript is the lack of comparison to the best solutions related to the FIRE dataset. On the  
https://projects.ics.forth.gr/cvrl/fire/
website you can find information about two publications:
1.  REMPE (H-M 17)
REMPE: Registration of Retinal Images through Eye Modelling and Pose Estimation
C. Hernandez-Matas, X. Zabulis, A.A. Argyros
IEEE Journal of Biomedical and Health Informatics, 2020
DOI: 10.1109/JBHI.2020.2984483 https://carlos.hernandez.im/papers/2020_04_JBHI.pdf

2. H-M 16
Retinal Image Registration Through Simultaneous Camera Pose and Eye Shape Estimation
C. Hernandez-Matas, X. Zabulis, A.A. Argyros
38th Annual International Conference of the IEEE Engineering in Medicine and Biology

Society (EMBC), pp. 3247-3251, Orlando, August 16-20, 2016
DOI: 10.1109/EMBC.2016.7591421

According to the information on the page, the AUC values achieved by the best algorithms are:  

                                        S     P     A     FIRE
1 REMPE (H-M 17)     0.958     0.542     0.660     0.773
2 H-M 16         0.945     0.443     0.577     0.721

The authors of the reviewed paper achieve lower AUC values. Therefore, it is difficult to indicate the advantages of the authors' proposal. If the authors demonstrate such advantages (speed, other datasets?) after comparing them to REMPE-type solutions (H-M 17), the article may be re-evaluated. Without such additions, the article should be rejected.

Round 2

Reviewer 1 Report

I would like to thank the authors for addressing my comments but please highlight your novelty and contribution by the end of the introduction section.

Reviewer 3 Report

I regret to say that the Authors did not respond to my comments in a factual manner.
1.
The above mentioned article [54] with definitely better results than the results in the reviewed article, was not included in the list of methods (Table 2 and Table 3). The authors in these tables only indicate selectively worse solutions, so as to indicate that their solution is the best.

2.
The article [54] has been marginalized by the authors of the reviewed article. The article [54] was only cited, but the results obtained were not given. It shoud be noted that the tests were carried out on the same FIRE datasete. Citing article [54] from 2020 together with articles [21] from 2006 and [55] from 1999, seems to be hide the advantages of article [54].

3.
The authors wrote in response to the reviewer:"However, with a more robust methodology for extracting bifurcations and crossovers, we anticipate that the results can be further improved." This comment is not backed by any research and has no justification.

Referring to my previous review, the authors did not improve the article, the results are worse than in [54], the advantages of the proposed solution were not demonstrated. The article should therefore be rejected.

Round 3

Reviewer 3 Report

The authors began to take into account my comments related to the need to cite and compare to article [54] - the REMPE method. It should be noted that article [54] is based on the FIRE database (also exclusively analyzed in the reviewed article). In addition, the authors of the article [54] provided software for the described registration process.

Comments:

1. Article [54] should be covered in the Related Work section, not just lines 444-449.Give the advantages and disadvantages (if any) of this solution.

2. Figure 12 must include the REMPE method as in article [54], where there is also a comparison to Harris-PIIFD and GDB-ICP (Figure 10 in [54]).

3. Table 3 does not include the REMPE method (why?). In my opinion, the reviewed article should include an additional table comparing the features of REMPE, Harris-PIIFD, GDB-ICP and the two proposed solutions. Such a table may indicate to the reader why the authors in the current Table 3 still do not include the REMPE solution.
